# The Relationship between Crawling and Emotion Discrimination in 9- to 10-Month-Old Infants

**DOI:** 10.3390/brainsci12040479

**Published:** 2022-04-05

**Authors:** Gloria Gehb, Michael Vesker, Bianca Jovanovic, Daniela Bahn, Christina Kauschke, Gudrun Schwarzer

**Affiliations:** 1Developmental Psychology, Justus Liebig University Giessen, 35394 Giessen, Germany; michael.vesker@gmail.com (M.V.); bianca.jovanovic@psychol.uni-giessen.de (B.J.); gudrun.schwarzer@psychol.uni-giessen.de (G.S.); 2Clinical Linguistics, Philipps University Marburg, 35032 Marburg, Germany; daniela.bahn@uni-marburg.de (D.B.); kauschke@staff.uni-marburg.de (C.K.)

**Keywords:** emotion discrimination, crawling, fear bias, morphed facial expressions

## Abstract

The present study examined whether infants’ crawling experience is related to their sensitivity to fearful emotional expressions. Twenty-nine 9- to 10-month-old infants were tested in a preferential looking task, in which they were presented with different pairs of animated faces on a screen displaying a 100% happy facial expression and morphed facial expressions containing varying degrees of fear and happiness. Regardless of their crawling experiences, all infants looked longer at more fearful faces. Additionally, infants with at least 6 weeks of crawling experience needed lower levels of fearfulness in the morphs in order to detect a change from a happy to a fearful face compared to those with less crawling experience. Thus, the crawling experience seems to increase infants’ sensitivity to fearfulness in faces.

## 1. Introduction

Emotional facial expressions play a crucial role in social interactions by allowing us to share our own feelings, as well as to evaluate the feelings of others. From birth, human beings are surrounded by emotional facial expressions, especially from their caregivers. These early interactions are characterized by positive facial expressions, and thus, during the first months of life, young infants prefer to look at positive faces, such as happy ones (*positivity bias*). This indicates that they are able to differentiate such positive expressions from negative expressions such as fear [1]. Interestingly, during the second half of the first year of life, there is a transition to preferring to look at negative over positive facial expressions, the so-called *negativity bias* [2]. Studies by Kotsoni et al. [3] and Cong et al. [4] on the categorical perception of happy and fearful faces were able to show just how fearful a facial expression needs to be in order to elicit an infant’s preference for looking at such faces over happy ones. However, one question that has not yet been sufficiently clarified is which factors influence infants’ sensitivity to and preference for fearful faces. A possible candidate could be the onset of infants’ self-produced locomotion, such as crawling, because the beginning of crawling brings many social-emotional changes, as well as changing the interactions between caregiver and child [5]. For instance, it was shown that there is an increase in expressions of anger and fear by the caregiver when infants start crawling [6,7]. In the present study, we wanted to investigate whether infants’ ability to crawl facilitates their ability to detect fear in a facial expression. In particular, we examined whether crawling infants are more sensitive to the differences between fearful expressions and happy expressions than non-crawling infants.

Perception and processing of emotional expressions in infancy: As mentioned above, during the first months of life, infants are usually surrounded by smiling, positive faces. Therefore, it is not surprising that studies showed that newborns prefer to look at happy faces [1], and this preference remains until 4 to 6 months of age [8]. By 5 months of age, infants are already able to distinguish between happy and fearful facial emotional expressions [9]. Other studies provide evidence that infants can also distinguish happy faces from other types of negative faces, such as sad and angry faces [10,11,12].

Studies also suggest a transition from a preference for looking at positive faces (*positivity bias*) to negative faces (*negativity bias*) during the first year of life [2]. In Ludemann and Nelson’s study [11], 7-month-olds looked longer at fearful faces compared to happy faces. Heck et al. [13] supported these findings by studying 3.5- and 5-month-olds. In this study, infants were presented with dynamic neutral, happy, and fearful faces. Five-month-old infants, but not 3-month-old infants, looked longer at the fearful faces than at neutral or happy faces. In a study by Quadrelli et al. [14], 7-month-olds showed a stronger neural response to happy faces compared to angry ones during a static presentation and a comparable neural response to angry and happy faces when the stimuli were presented dynamically. However, when happy and angry facial expressions were presented statically in a study by Grossmann et al. [15], 7-month-old infants showed a higher sensitivity to happy faces than to angry faces. Despite these somewhat heterogeneous findings regarding the exact time point of the transition from a *Positivity bias* to a *Negativity bias*, in general, it seems that regarding the contrast between happy and fearful facial expressions, this transition seems to manifest at the latest around 7 months of age. However, it is important to note that the above studies compared prototypical pure expressions of these emotions.

Intensities required to detect changes in emotional facial expressions in infancy: In order to react quickly and appropriately in certain situations (particularly in dangerous or ambiguous situations), it can be important to recognize positive or negative emotional information in facial expressions on a finer-grain level beyond the prototypical posed emotional expressions. In order to investigate how fearful an expression needs to be in order for infants to detect the transition from a happy to a fearful facial expression, Kotsoni and colleagues [3] examined 7 months old infants using a preferential looking task. They created a morphed continuum from 100% happy to 100% fearful facial expressions in 20% increments. In total, they used six different images of the facial expressions and presented them in pairs: a 100% happy face was always paired with one of the other faces from the happy–fearful continuum. They found that starting from the 60% fearfulness/40% happiness morph onward, infants looked longer at the fearful faces as the morphs became more fearful. These findings suggest that 60% fearfulness could be the boundary indicating the amount of fear-related information that infants at this age require to notice the difference between happy and fearful faces. Cong and colleagues [4] replicated Kotsoni et al.’s [3] study conceptually and also examined adults for comparison. As in the previous study, infants in Cong et al.’s [4] study also showed a fear preference starting from the 60% fearfulness morph. However, the adults became sensitive to the distinction between happy and fearful morphs, starting from only 30–40% fearfulness. The lower quantity of visual information indicating a fearful facial expression required by adults to distinguish happiness from fear as compared to infants suggests that they are more sensitive to fearful facial expressions. This increased sensitivity could at least partially stem from adults having more experience with such faces over the course of their lives, which makes sense given that the identification of fear or anger in facial expressions can be especially relevant for survival. For example, studies have shown that adults’ attention is better captured by negative faces than by positive or neutral ones [16,17]. A distinct point in time when infants could begin to have increased exposure to such faces is when they begin to crawl, as will be further explored below.

Crawling and emotion processing: One of the most important milestones during the first year of life is the onset of crawling. In the second half of the first year of life, most infants start to locomote on their hands and knees. As soon as infants start to locomote on their own, there are remarkable changes in perceptual and cognitive skills, as well as social-emotional development. For an overview, see [5,18].

For instance, Campos [19] found that crawling increased the infants’ own expression of positive and negative emotions, as well as increased social referencing in unfamiliar situations (i.e., referring to the reactions of others to inform their own responses to situations). Social referencing emerges in the second half of the first year of life [20] and forms the basis for social learning and social appraisal in adulthood [21]. Prior studies showed that even young infants use the emotional expressions of others to understand the meaning of ambiguous situations, e.g., for review, [2,21,22,23]. In these studies, infants guided their own behavior based on their caregivers’ behavior. If the parents showed positive emotions in a situation, the infants were more willing to perform certain actions. However, if the parents showed negative emotions, the infants did not show approach behavior. A popular example of social referencing in infancy is the willingness of infants to cross an apparent visual cliff. Sorce et al. [22] used such an apparatus to study crawling infants: If the mother was encouraging and happy, the majority of infants crossed the visual cliff. However, if the mother expressed fear and anger, only a few infants were willing to cross the visual cliff. Furthermore, infants in a study by Vaish and Striano [24] crossed the visual cliff faster when mothers showed facial and vocal cues simultaneously compared to only vocal cues. The importance of social referencing regarding locomotion in uncertain situations was also shown in older infants. Karasik et al. [25] examined 12-month-old experienced crawlers in an adjustable visual cliff task. The landing platform was adjustable in 1-cm-increments (drop-off range from 0 cm to 90 cm). The mothers stood at the end of the platform and either encouraged or discouraged their babies in a natural way to cross the cliff regardless of the depth. Particularly in cases when perceptual information was ambiguous (“risky cliffs”), experienced crawlers deferred to social information, i.e., when mothers discouraged their infants, they did not cross the ambiguous cliff and vice versa. Eighteen-month-olds were also shown to use social referencing when they had to cross an ambiguous slippery slope [26,27]. These kinds of situations show the importance of early recognition of emotional cues since they are essential for infants to guide their behavior based on the expressions of their caregivers and thereby help to protect them from potential injuries. More specifically, negative emotions (especially fear) are particularly informative for infants to avoid potential danger, which could also be why infants seem to pay more attention to fearful faces in the second half of the first year of life [2].

There is also evidence to suggest that adults naturally regulate infants’ behavior by mimicry or verbal responses [5]. For example, in Tamis-LeMonda et al.’s [28] study, mothers reacted in unexpected dangerous situations by prohibiting words accompanied by negative (fearful) facial expressions or showing more anger [6,7]. In summary, these findings suggest that when infants start crawling, they start to pay more attention to their caregivers’ reactions as the caregivers begin to show more negative emotional expressions to protect their infants from danger. This is not surprising since crawling infants could encounter dangerous situations more often than non-crawling infants and are thus more dependent on the facial expressions of others for survival. We hypothesized that this increased experience of crawling infants with negative expressions could increase their sensitivity to negative emotions compared to non-crawling infants.

The current study: The current study was aimed at filling the gap in the research mentioned above by investigating whether self-produced locomotion ability in the form of crawling influences infants’ ability to detect fear in facial expressions. To this end, we created a set of morphed emotional facial expressions on a continuum from 100% happy to 100% fearful of finding out at which point of the continuum infants begin to respond to the fearfulness conveyed by the morphed faces. Additionally, we asked the caregivers about the infants’ crawling experience using the German version of the Bayley III scales [29] and specifically asked about the exact time of the infants’ onset of crawling. Furthermore, we asked the parents about their stress levels to determine whether the parents of crawling infants feel more stressed than parents of non-crawling infants. In order to exclude the possibility that crawling and non-crawling infants might differ in their general emotional status, a questionnaire regarding this topic was also given to the caregivers.

Given previous studies, e.g., [9], we expected that all infants, regardless of their crawling experience, would show a preference for looking at the more fearful facial expressions (*fear bias*). Furthermore, due to the links between crawling and infants’ increased experience with negative emotional expressions reported above, we hypothesized that crawling infants would begin to show a looking preference for fearful expressions (versus happy expressions) at a lower degree of fearfulness compared to non-crawling infants.

Since the onset of crawling brings many changes in social interaction [5], and caregivers begin to show more negative emotions towards their infants [6,7], we expected that parents of crawling infants might have a higher stress level than parents of non-crawling infants. Furthermore, we expected that infants of parents with higher stress levels might show a higher preference for fearful faces compared to infants of parents with lower stress levels.

## 2. Materials and Methods

Ethical statement: The current study was conducted in accordance with the German Psychological Society (DGPs) Research Ethics Guidelines. The Office of Research Ethics of the Justus-Liebig-University Giessen approved the experimental procedure and the informed consent protocol. Prior to participation in the study, written informed consent was obtained from the parents of the infants.

Participants: Infants were recruited by obtaining birth records from local municipal councils and neighboring communities, as well as through personal recruitment at the obstetrics department of a cooperating hospital and infant-care courses. The final sample consisted of 29 healthy full-term infants (8 female and 6 male crawlers (crawling duration: *M* = 71.93 days, *SD* = 14.96 days); 3 female and 12 male non-crawlers (crawling duration: *M* = 7.67 days, *SD* = 11.01 days)) with a mean age of 9 months and 27 days (*SD* = 15 days). Fourteen further infants were tested but excluded from our data analyses because of crying and discomfort during testing (*n* = 3), missing eye-tracking data (*n* = 3), experimental errors (*n* = 2), errors in the eye-tracking program (*n* = 1), and an insufficient amount of gaze data (*n* = 5). During the test appointment, caregivers were informed about the study procedure by the experimenter and signed an informed consent form. Recruited infants were predominantly of Caucasian background and lived in Giessen and suburban areas of Giessen.

### 2.1. Materials and Stimuli

Emotion task: The original photographs (100% happy and 100% fearful faces) were obtained from the McGill University Pell Laboratory database and were used for the current study with the consent of the responsible parties [30]. The morphed stimuli were created using the FantaMorph 5 morphing software package [31]. Two photographs of the same Caucasian woman displaying a 100% happy face and a 100% fearful face served as the template to create the morphed image continuum. By combining the two initial images using the morphing software with each image contributing a varying degree of information, we produced a series of 11 intermediate morphs in 10% increments ranging from 100% happy/0% fearful to 0% happy/100% fearful (Figure 1). Paint.net software [32] was then used to remove artifacts (e.g., teeth that were too dark) from the morphed images. The luminance for each image was obtained using GIMP version 2.10.12 [33].

In order to make the presentation of the morphed expressions more natural and dynamic, animations from our series of still morphs were generated, once again using FantaMorph 5 [31]. The animations depicted a continuous formation of each morphed face as well as the 100% happy face over the course of 500 ms (15 frames) from a neutral facial expression (for an example, see Figure 2).

When appearing on the presentation screen, each stimulus had a height of 16.8 cm and a width of 11.4 cm with a resolution of 412 × 604 pixels (see Figure 3).

Parents’ stress level: In order to obtain information about the current mental state of the parents, we used a state of mind questionnaire [34]. The state of mind questionnaire includes 24 items about the current well-being of the respondent. The raw values obtained are converted into T-values (or PRs or stanine values) for subsequent interpretation. Thus, a T-score above 60 is considered slightly elevated, a T-score ≥63 is considered moderately elevated, and a T-score ≥70 is considered significantly elevated. In the opposite case, T values ≤40 are considered very low [34]. Additionally, the German version of the Recovery-stress questionnaire [35] was used to obtain information regarding the parents’ current extent of recovery and stress over the previous three days and nights. The basic version with 24 items was used [35].

Infants’ social-emotional development level: In order to assess the infants’ general level of social-emotional development, we used age-specific emotion-related items from a social-emotion questionnaire [36] administered to the parents. The questions referred to self-image, emotional independence, awareness of reality, moral development, anxiety, impulse control, and regulation of emotions.

Infants’ crawling status: In order to determine the infants’ crawling status, parents were asked about their infants’ crawling status based on the definition of crawling in the German version of The Bayley Scales of Infant and Toddler Development-III [29]. Crawling was defined as follows: “Child makes forward progress of at least 1.5 m by crawling on hands and knees”.

Background questionnaire: A custom questionnaire was used to record socio-economic status, number of siblings, and duration of pregnancy.

### 2.2. Apparatus and Procedure

Emotion task: Stimuli were presented using E-Prime 3 [37] on an LCD monitor (diagonal size: 61 cm) with a resolution of 1920 × 1080 pixels. An eye-tracker was attached below the screen (Tobii Pro X3-120, Stockholm, Sweden). The eye movements and gaze durations were recorded at 120 Hz using Tobii-Studio 3.4.7 [38]. Fixations were identified using the *ClearView Fixation Filter* with a velocity threshold of 50 pixels/sample and a duration threshold of 100 ms. The area inside the gray frame of each stimulus was set as the *area of interest* (AOI).

Infants sat on their caregiver’s lap, who was seated on a chair positioned such that the infant’s head was approximately 60 cm from the screen. The chair was adjusted in height to ensure that infants’ eyes were lined up with the middle of the screen. In order to minimize visual distractions, barriers were placed behind the screen, as well as on the left and right sides of the testing area. Before the experiment started, parents were asked to wear sunglasses and close their eyes (if possible) to ensure that only the infants’ gaze was tracked. The experiment started with a 5-point-calibration (2-point if the 5-point calibration was not successful after two attempts), where the infants’ attention was attracted to the calibration points on the screen by animated animals. After the calibration, 20 experimental trials were presented on the screen, separated by a colorful rotating attention-getter (with the addition of an auditory signal to attract the infants’ attention). As soon as the infant looked at the attention-getter, the experimenter activated the next trial, which began with an auditory clip of a bell, and a hash symbol appearing at the center of the screen for 1 s. If the infant did not look at the attention-getter after 10 rotations, the trial began automatically. Once the hash symbol disappeared, a pair of facial animations were presented for 10 s. The pairs always consisted of the 100% happy face animation and one of the happy–fearful animated morphs. All 10 possible pair combinations were presented, with each particular pair appearing twice, once with the 100% happy animation on the left and the happy–fearful morph on the right, and the second time with the 100% happy animation on the right and the happy–fearful morph on the left (see Figure 3), for a total of 20 trials. The presentation order was randomized and divided into two blocks: in one block, infants saw the 10%, 30%, 50%, 70%, and 90% happy–fearful morphs (each paired with the 100% happy face), and in the other block infants saw the 20%, 40%, 60%, 80%, and 100% happy–fearful morphs (each paired with the 100% happy face) to ensure that successive trials showed morphs separated by at least two 10% increments. This arrangement helped to avoid the risk of infants seeing a strong concentration of morphs from either end of the continuum at some point in the course of the experiment through chance. The order of the blocks was randomized across participants.

Questionnaires: After the main experimental task, caregivers were asked to fill out the questionnaires.

Dependent variable in the emotion task: Since Tobii studio’s fixation filter was set to 100 ms, a fixation was only recorded for analysis if it lasted at least 100 ms. Furthermore, only fixations within the defined AOIs were recorded for analysis. For the looking time measure, we used the total fixation durations for each animated expression from each trial, i.e., the total time during each trial the infant looked towards each stimulus (100% happy face and morphed face). For each of the 10 morph levels, we averaged the looking times across the two trials, which showed the morph on either the left or the right side of the screen. We then calculated looking preference scores as follows for each infant:$$\mathrm{preference}\;\mathrm{score}=\left(\frac{\mathrm{looking}\;\mathrm{time}\;\mathrm{to}\;\mathrm{the}\;\mathrm{morphed}\;\mathrm{face}}{\left(\mathrm{looking}\;\mathrm{time}\;\mathrm{to}\;\mathrm{the}\;\mathrm{morphed}\;\mathrm{face}\;+\;\mathrm{looking}\;\mathrm{time}\;\mathrm{to}\;\mathrm{the}\;100\%\;\mathrm{happy}\;\mathrm{face}\right)}\right)\times 100$$

## 3. Results

To analyze the data, we used SPSS 27 [39]. First, we analyzed the differences between crawling and non-crawling infants in terms of their mean social-emotional age, the number of siblings, parents’ educational level, parents’ stress level (Bf-SR-T-score; [34]), as well as parents’ overall stress and recovery level (EBF; [35]). Therefore, we performed univariate ANOVAs on these variables using crawling status (crawler vs. non-crawlers) as a between-subjects factor. These analyses revealed no significant results (all *p*s ≥ 0.128), and thus these factors were not included in our further analyses of preference scores regarding the emotional task. The descriptive values are shown in Table 1.

In order to analyze whether infants’ preference scores were influenced by the morph level and their crawling status, we conducted a repeated-measures ANOVA on the preference scores using the morph level (10% to 100%) as a within-subject factor and the crawling status (crawler vs. non-crawler) as a between-subjects factor. The results showed a significant effect of morph level, *F*(9, 243) = 2.355, *p* = 0.014, η^2^_part_ = 0.080, but no interaction with crawling status, *F*(9, 243) = 1.655, *p* = 0.101, η^2^_part_ = 0.058, and no main effect of crawling status, *F*(1, 27) = 0.372, *p* = 0.547, η^2^_part_ = 0.014. The mean preference scores at each morph level for all infants, regardless of crawling status, are depicted in Figure 4. A post hoc power analysis revealed a power of 0.99.

In order to further analyze the increase in preference scores with increasing morph level, which we observed in the previous analysis at approximately the midpoint of the continuum (see Figure 4), we compared the average preference scores from the first half of the continuum (10% to 50%) against the average preference scores from the second half of the continuum (60% to 100%). This analysis also allowed us to examine whether the infants in the current study show a general fear bias, as shown by previous studies that investigated similar age groups of infants using fearful and happy facial stimuli [3,4]. We again ran a repeated-measures ANOVA on the preference scores with the morphing degree (10% to 50% vs. 60% to 100%, as described above) as a within-subject factor and crawling status (crawlers vs. non-crawlers) as a between-subjects factor. The results showed a significant effect of the morphing degree, *F*(1, 27) = 7.797, *p* = 0.009, η^2^_part_ = 0.224, with infants showing higher preference scores for morphs from the second half of the continuum (Figure 5). There was no significant interaction between the morphing degree and crawling status, *F*(1, 27) = 2.092, *p* = 0.160, η^2^_part_ = 0.072, and no main effect of crawling status, *F*(1, 27) = 0.372, *p* = 0.547, η^2^_part_ = 0.014.

Thus, as depicted in Figure 5, we found that from morph level 60% onward, all infants showed a significant looking preference for fearful morphs compared to a happy face.

This result is similar to our findings from the first analysis, which suggested a change in looking preferences occurring between 40% and 50% of the fearful morph continuum (Figure 4), and seemed to be very close to the 60% fear threshold found by Cong et al. [4] and Kotsoni et al. [3] in infants. We, therefore, hypothesized that the 40% to 60% morphing range might represent a crucial sensitivity range in which infants at this age begin to distinguish between fearful and happy expressions. We then further hypothesized that if the effect of crawling ability was a relatively weak one, then it might be most influential and detectable within this range of 40% to 60% of the morphing continuum where the stimuli are most ambiguous.

In order to test this hypothesis, we carried out an additional repeated-measures ANOVA focused on only the preference scores for the 40%, 50%, and 60% morphs. Once again, morph level (40%, 50%, 60%) served as a within-subject factor and the crawling status (crawler vs. non-crawler) as a between-subjects factor. The results showed a significant effect of morph level, *F*(2, 54) = 3.783, *p* = 0.029, η^2^_part_ = 0.123, as well as a significant interaction between the morph level and the crawling status, *F*(2, 54) = 3.734, *p* = 0.030, η^2^_part_ = 0.121, but no significant main effect of crawling status, *F*(1, 27) = 2.631, *p* = 0.116, η^2^_part_ = 0.089. To further analyze the interaction between morph level and crawling status (Figure 6), we conducted separate post hoc univariate ANOVAs for each morph level (40%, 50%, and 60%) on the preference scores, with crawling status serving as a between-subjects factor.

The results at 40% showed a marginally significant, *F*(1, 27) = 4.051, *p* = 0.054, η^2^_part_ = 0.130, trend of the preference scores differing between the crawlers (*M* = 51.59%, *SD* = 15.27%) compared to the non-crawlers (*M* = 40.81%, *SD* = 13.57%), suggesting a slight preference for the fearful morph in the crawlers, and a preference for the happy face in the non-crawlers.

At the 50% morph level, there was a significant difference between crawling and non-crawling infants, *F*(1, 27) = 4.823, *p* = 0.037, η^2^_part_ = 0.152. Crawlers showed noticeably higher preference scores for the morphed face (*M* = 61.14%, *SD* = 16.54%) than non-crawlers (*M* = 49.73%, *SD* = 11.08%). Therefore, at 50% of the fearful morphing continuum, only the crawlers were sensitive to fearful information in the morphed facial expression.

At 60%, we found no significant difference between crawlers and non-crawlers, *F*(1, 27) = 1.779, *p* = 0.193, η^2^_part_ = 0.062. Here, both the crawlers (*M* = 52.08%, *SD* = 15.25%) and non-crawlers (*M* = 59.90%, *SD* = 16.27%) showed preference scores over 50%, suggesting that from this point onwards, the morphs resembled the fearful expression enough that both groups showed a looking preference for the fearful morph over the 100% happy expression. See Figure 6 for an illustration of the differences between crawling and non-crawling infants at the 40%, 50%, and 60% levels of the fearful morph continuum.

Three additional repeated-measures ANOVAs were then carried out to check for the influence of parents’ stress and recovery levels on the infants’ looking preference scores, which served as the dependent variable in all three analyses. In the first analysis, morph level (10% to 100% fear) was used as a within-subject factor, with the EBF-stress-group (more stressed, *n* = 15 vs. less stressed *n* = 14; median split) as a between-subjects factor, and the EBF-overall-stress-score as a continuous covariate. No significant main effects or significant interactions were revealed (all *p*s ≥ 0.161). The second analysis was carried out with morph level (10% to 100% fearful facial expression) as a within-subject factor, EBF-recovery-group (more recovered, *n* = 16 vs. less recovered *n* = 13; median split) as a between-subjects factor, and the EBF-overall-recovery-score as a continuous covariate. This analysis did not show any significant results (all *p*s ≥ 0.058). Thus, neither parents’ stress level nor parents’ recovery level had a significant influence on infants’ preference scores. The third repeated-measures ANOVA was also carried out with morph level (10% to 100% fear) as a within-subject factor, the Bf-SR group (more stressed, *n* = 16 vs. less stressed *n* = 13; median split) as a between-subjects factor, and the Bf-SR-T score as a continuous covariate. Once again, there were no significant main effects or significant interactions (all *p*s ≥ 0.102), indicating that this parental stress score was not related to infants’ looking preference scores. In order to obtain more insight into the parents’ stress levels, we performed one final analysis based on the interpretation of the Bf-SR [34]. Based on the Bf-SR-T scores parents were assigned to four groups (strikingly low: *n* = 3; normal: *n* = 23; slightly elevated: *n* = 2; markedly elevated: *n* =1). We then ran a χ^2^-test to check whether these groups were equally distributed. As expected from the group sizes, the groups were not equally distributed, χ^2^(3) = 45.897, *p* < 0.001, with most parents showing normal stress levels.

## 4. Discussion

The main motivation for the present study was to investigate whether infants’ self-produced locomotion influences their ability to detect fear in facial expressions. In particular, we wanted to know whether crawling infants would begin to show a looking preference for fearful expressions over happy expressions at a lower degree of fearfulness compared to same-aged non-crawling infants.

First, our results showed that the degree of fearfulness in the morphed faces significantly influenced infants’ looking behavior. In a statistical analysis in which all face pairs were included, we found that regardless of their crawling status, infants began to show a looking preference for the fearful morphs over the happy face starting from the 60% morph onward as the faces became more fearful. This result confirms previous findings, indicating a so-called *negativity bias*, e.g., [2], or more specifically, a *fear bias*, e.g., [3,4] in the second half of the first year of life.

However, this analysis did not show a significant influence of crawling on the infants’ preference scores, suggesting that any such effect might be relatively weak. We, therefore, hypothesized that crawling might have the strongest influence on looking behavior (and thus be more detectable) at or around the 50% fear morph level since this is where the morphs are the most ambiguous. We reasoned that in morphs below 50% fear, the fearful expressions may have still been too similar to the happy face to be easily distinguishable from it by either crawling or non-crawling infants. Following the same logic, morphs above 50% may have appeared to be distinct enough from the happy face that both crawlers and non-crawlers could reliably differentiate between them. Thus, we reasoned that around 50% of the morphs could be ambiguous enough for even a relatively weak effect of crawling to give crawling infants a noticeable advantage in distinguishing the fearful morph from the happy face. In order to test this hypothesis, we carried out a further analysis comparing the crawlers and non-crawlers, this time focusing specifically on the halfway point of the morphing continuum (50%), as well as the two neighboring morph levels (40% and 60%) for comparison, but without the other morph levels to avoid diluting the statistical power of the analysis. Our results from this focused analysis showed a significant interaction between the degree of morphing and the infants’ crawling status, with crawling infants showing significantly higher looking preference scores for the morph over the happy face, but only at a 50% morphing level. This result confirmed our hypothesis that the effect of crawling is strongest at the most ambiguous 50% morph. Thus, in the current study, infants showed an overall transition to a *fear bias* at around 50% of the fearful expression, and at this transition point, the crawling infants appeared to be more sensitive to fearful facial expressions than same-aged non-crawlers. In other words, when infants are examined in terms of their self-locomotion experience, crawlers seem to undergo the transition to a *fear bias* at 50% of the morphing continuum, whereas non-crawlers do not show this transition until 60%. When one compares our results to earlier studies that used a similar approach, the sensitivity of non-crawling infants to fearful faces in our study was similar to the younger 7-month-old infants in the studies by Kotsoni et al. [3] and Cong et al. [4], were the infants also showed a looking preference transition at 60% of the fearful expression. Meanwhile, the 50% transition point we saw in crawling infants in our study was closer to the 40% transition point that was seen in adult participants by Cong et al. [4]. Therefore, it appears that the potential contribution of crawling to the development of infants’ sensitivity to fearful faces fits within the broader development of this sensitivity from infancy to adulthood.

As to why crawling infants should be more sensitive to fear, one possible explanation is that crawling infants can encounter more potentially dangerous situations than infants who are not yet able to crawl. Therefore, crawling infants would have more experience in using social referencing and detecting signals of danger in the faces of their interaction partners, e.g., [22,24], in order to evaluate the safety of their movements. Examples of such effects include studies where experienced crawlers avoided crossing a visual cliff if their mothers expressed fear [22].

Our results on the effect of crawling with regards to more ambiguous morphs appear to broadly agree with studies showing that the use of social referencing by infants to evaluate a situation and guide their own behavior seems to be particularly relevant in ambiguous situations. For instance, in Karasik et al.’s [25] study, infants were more dependent on their caregivers’ encouraging or discouraging behavior in deciding whether or not to cross an ambiguous visual cliff. If the caregivers showed discouraging behavior, the infants did not cross the ambiguous cliff and vice versa. By contrast, when the visual cliff appeared to be safe, infants crossed the cliff independent of their caregivers’ social cues and struggled to cross the cliff if it appeared to be dangerous regardless of caregivers’ reactions.

An important point to note is our use of dynamic stimuli, which can introduce low-level motion-related differences between the visual stimuli. This question is quite important since some studies have demonstrated that infants show more attention to moving stimuli compared to static stimuli, e.g., [40]. This point was raised by Grossmann and Jessen [41] in reference to another study [13] that used dynamic stimuli. Grossmann and Jessen [41] noted that in this study [13], the fearful facial expressions contained more movement than the happy and neutral facial expressions. However, the happy animated face in our study was only used as a contrast against which to measure the infants’ sensitivity to the fearful morphs, which was the real variable of interest. Furthermore, to test our hypothesis regarding the effects of crawling, we used a between-subjects design where all infants saw the same selection of stimuli, and thus any low-level visual differences between the happy and fearful faces are very unlikely to impact our findings significantly.

## 5. Conclusions

Independent of infants’ crawling experience, 9- to 10-month-old infants were able to detect a change from a happy to a fearful facial expression starting from the 60% fearful morph level. Furthermore, our study showed that 9- to 10-month-old experienced crawlers are more sensitive to fearfulness in faces than same-aged non-crawling infants: Crawling infants were able to differentiate happy faces from fearful morphs starting from morphs containing only 50% of the fearful expression. We propose that this advantage of crawling infants with respect to perceiving fearful faces may be caused by their higher familiarity with fearful expressions due to their use of social referencing [22,24,25] as caregivers provide cautionary feedback when they move about independently [6,7,28].

## 6. Limitations and Future Research

One noteworthy limitation of our study is that infants were presented with unfamiliar faces. Considering that our central hypothesis relies on infants gaining experience with processing fearful facial expressions based on interactions with their caregivers, an interesting follow-up question is whether the infants’ processing of fearful expressions produced by their own caregivers would follow the same pattern we observed in the present study. Furthermore, it would be interesting to observe the degree to which their processing ability for fearful faces translates to action. For instance, a visual cliff task could be carried out with infants observing facial feedback from either their own caregiver or a stranger. Additionally, the infants in this task could be presented with morphed facial expressions based on a stranger’s face or the face of their caregiver. Such a study design could provide insight into how strong an emotional facial expression needs to be in order to influence infant behavior in terms of both gaze and decision making, as well as clarify the role of familiarity in this relationship. Such studies could also examine infants’ looking behavior with respect to individual parts of the face in order to determine their role in the infants’ perception of emotional facial expressions.

Another question that we could not reliably answer in our study is the influence of parental stress level on the infants’ sensitivity to fearful faces since the current study involved almost exclusively (88.5%) parents with normal stress levels. Future studies could address this point with a broader selection of study participants, which could be especially valuable from a clinical standpoint.

## Figures and Tables

**Figure 1 brainsci-12-00479-f001:**
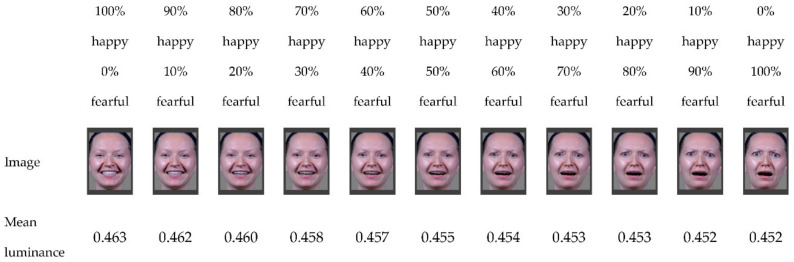
Continuum between 100% happy and 100% fearful facial expressions in 10% increments. Mean luminance (from 0.000 to 1.000) for each image was obtained using GIMP version 2.10.12 [33].

**Figure 2 brainsci-12-00479-f002:**
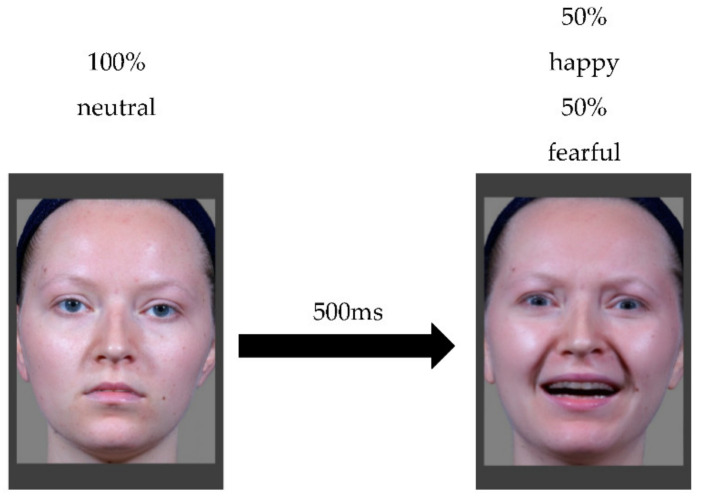
Example of a dynamic morphed stimulus animation.

**Figure 3 brainsci-12-00479-f003:**
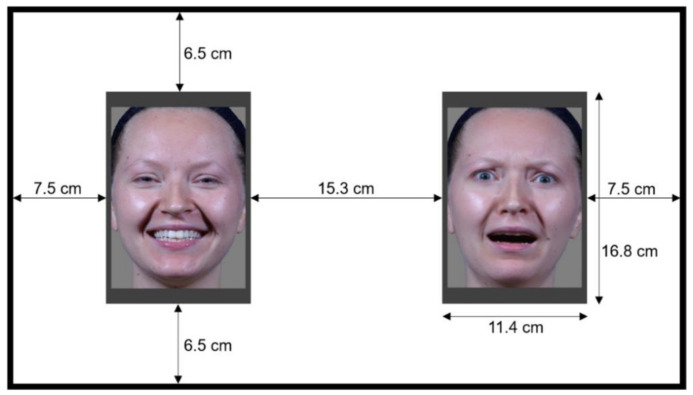
The dimensions of stimuli in relation to the screen.

**Figure 4 brainsci-12-00479-f004:**
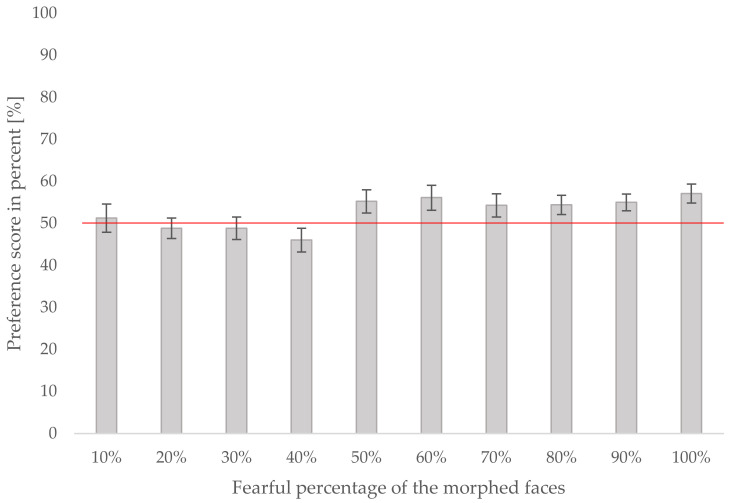
The preference scores at each morph level (10% to 100% of fearful facial expression) in all infants regardless of crawling ability. Error bars indicate the standard error of the mean.

**Figure 5 brainsci-12-00479-f005:**
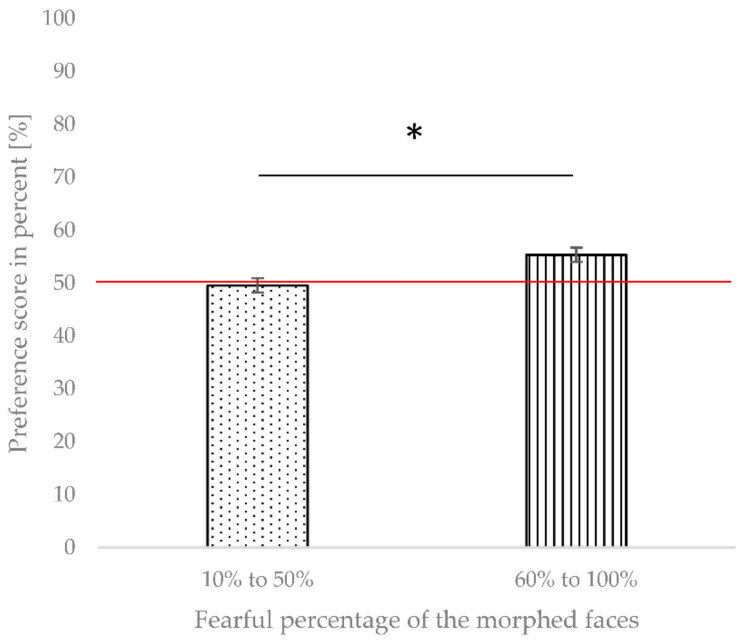
Comparison of preference scores for the first half of the morphing continuum (10% to 50% fearful facial expression) versus preference scores for the second half of the morphing continuum (60% to 100% fearful facial expression) for all infants regardless of crawling ability. Error bars indicate the standard error of the mean. * indicates a *p*-value < 0.05.

**Figure 6 brainsci-12-00479-f006:**
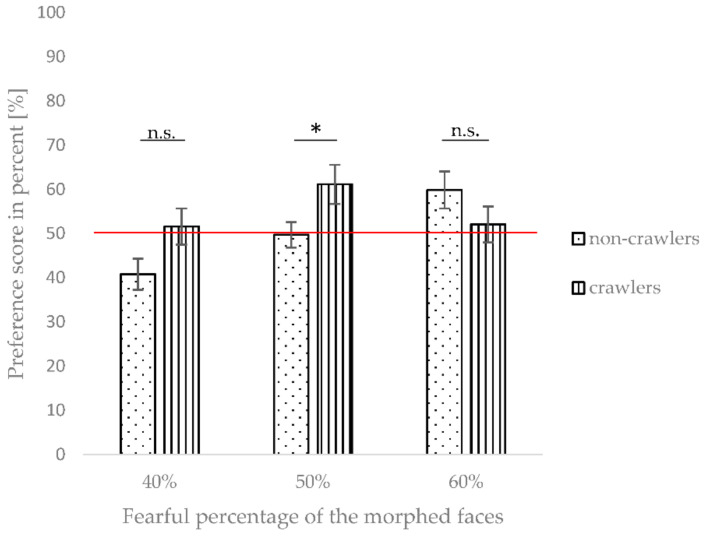
The preference scores for 40%, 50%, and 60% fearful morph levels for crawling and non-crawling infants. Error bars indicate the standard error of the mean. * indicates a *p*-value < 0.05, and n.s. (non-significant) indicates *p*-values > 0.05.

**Table 1 brainsci-12-00479-t001:** Means and standard deviations of the different questionnaires.

Questionnaire Variables	Crawlers (*n* = 14)	Non-Crawlers (*n* = 15)
Number of siblings	*M* = 0.64; *SD* = 0.93	*M* = 0.40; *SD* = 0.63
Mothers’ educational level	*M* = 3.38; *SD* = 0.87	*M* = 3.27; *SD* = 0.88
Fathers’ educational level	*M* = 2.92; *SD* = 1.38	*M* = 3.13; *SD* = 0.99
Infants’ social-emotional age	*M* = 1.18; *SD* = 0.21	*M* = 1.08; *SD* = 0.24
Parents’ stress level (Bf-SR-T-score)	*M* = 49.64; *SD* = 6.97	*M* = 51.47; *SD* = 7.99
Parents’ overall stress and recovery level (EBF)	*M* = 1.56; *SD* = 0.59 *M* = 2.82; *SD* = 0.79	*M* = 1.42; *SD* = 0.61 *M* = 2.78; *SD* = 0.70

*Note.* Mothers’ and fathers’ educational levels ranged from 0 (no degree) to 5 (doctorate/habilitation). For one infant in the crawling group, the parents did not answer the questions regarding the mother’s and father’s educational level.

## Data Availability

Data used for our analyses are made available on Zenodo under the title of this publication.

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
