# Peer review of "The Relationship between Crawling and Emotion Discrimination in 9- to 10-Month-Old Infants"

_brainsci, 2022, doi:10.3390/brainsci12040479_

Round 1

Reviewer 1 Report

The manuscript is very important. It provides an investigation of the relation between motor and cognitive development. The manuscript is informative and well-written. 
Please find my follow comments in order to improve the quality of the manuscript: 
Abstract: The conclusion is not related to study aim or results. 
Introduction: - Please rewrite line 44
- Please short the paragraph from line 62 to 93 
- Please discuss the mechanism of the relation between motor ability or milestone, and facial expression recognition
Discussion: 
- There is no explanation of the study results 
- There is no sufficient comparison with the current evidence 
- There is no research nor clinical recommendations 

Author Response

The manuscript is very important. It provides an investigation of the relation between motor and cognitive development. The manuscript is informative and well-written. Please find my follow comments in order to improve the quality of the manuscript: Abstract: The conclusion is not related to study aim or results.
ï‚· We now substantiated our conclusion with related studies as follows (see line 474 to 483):
o Conclusion: Independent of infants’ crawling experience, 9- to 10- months-old infants were able to detect a change from a happy to a fearful facial expression starting from the 60% fearful expression morph level. Furthermore, our study showed that 9- to 10- month-old experienced crawlers are more sensitive to fearfulness in faces than same-aged non-crawling infants: Crawling infants were able to differentiate happy faces from fearful morphs starting from morphs containing only 50% of the fearful expression. We propose that this advantage of crawling infants with respect to perceiving fearful faces may be caused by their higher familiarity with fearful expressions due to their use of social referencing [22,24,25] as caregivers provide cautionary feedback when they move about independently [6,7,28].
ï‚· Based on this conclusion, we changed our abstract (see line 11 to 19)
o Abstract: The present study examined whether infants' crawling experience is related to their sensitivity to fearful emotional expressions. Twenty-nine 9- to 10-month-old infants were tested in a preferential looking task, in which they were
presented with different pairs of animated faces on a screen displaying a 100% happy facial expression and morphed facial expressions containing varying degrees of fear and happiness. Regardless of their crawling experiences all infants looked longer at more fearful faces. Additionally, infants with at least 6 weeks of crawling experience needed lower levels of fearfulness in the morphs in order to detect a change from a happy to a fearful face compared to those with less crawling experience. Thus, crawling experience seems to increase infants’ sensitivity to fearfulness in faces.
Introduction:
- Please rewrite line 44
ï‚· Thank you for the advice. We rewrote the sentence as follows (see line 46 to 48):
o Perception and processing of emotional expressions in infancy: As mentioned above, during the first months of life, infants are usually surrounded by smiling positive faces. Therefore, it is not surprising that studies showed that newborns prefer to look at happy faces [1], and this preference remains until 4 to 6 months of age [8].
- Please short the paragraph from line 62 to 93
ï‚· We have shortened the paragraph (see line 71 to 98) regarding the differences during their habituation in Kotsoni et al.’s 2001 and Cong et al’s 2009 studies. On the other hand, we had to add two further sentences regarding adults’ behavior. Please, see below:
o Intensities required to detect changes in emotional facial expressions in infancy: In order to react quickly and appropriately in certain situations (particularly in dangerous or ambiguous situations) it can be important to recognize positive or negative emotional information in a facial expression on a finer-grain level beyond the prototypical posed emotional expressions. To investigate how fearful an expression needs to be in order for infants to detect the transition from a happy to a fearful facial expression, Kotsoni and colleagues [3] examined 7 months old infants using a preferential looking task. They created a morphed continuum from 100% happy to 100% fearful facial expressions in 20% increments. In total, they used six different images of the facial expressions and presented them in pairs: a 100% happy face was always paired to one of the other faces from the happy-fearful continuum. They found that starting from the 60% fearfulness/40% happiness morph onward, infants looked longer at the fearful faces as the morphs became more fearful. These findings suggest that 60% fearfulness could be the boundary indicating the amount of fear-related information that infants at this age require to notice the difference between happy and fearful faces. Cong and colleagues [4] replicated Kotsoni et al.’s [3] study conceptually, and also examined adults for comparison. As in the previous study, infants in Cong et al.’s [4] study also showed a fear preference starting from the 60% fearfulness morph. However, the adults became sensitive to the distinction
between happy and fearful morphs starting from only 30 – 40% fearfulness. The lower quantity of visual information indicating a fearful facial expression required by adults to distinguish happiness from fear as compared to infants suggests that they are more sensitive to fearful facial expressions. This increased sensitivity seems likely to at least partially stem from adults having more experiences with such faces over the course of their lives, which makes sense given that the identification of fear or anger in facial expression can be especially relevant for survival. For example, studies have shown that adults’ attention was better captured by negative faces than by positive or neutral ones [16,17]. A distinct point in time when infants could begin to have increased exposure to such faces is when they begin to crawl, as will be further explored below.
- Please discuss the mechanism of the relation between motor ability or milestone, and facial expression recognition
ï‚· We added more discussion regarding these relations in the crawling and emotion processing section of the introduction. There are changes in lines 100-101; 132; and 142-144
Discussion: - There is no explanation of the study results
ï‚· We added the Kotsoni et al. 2001 and Cong et al. 2019 studies to explain the weakness of the effect of crawling in our analyses. Please see (line 390 to 406 in the manuscript) below:
o First, our results showed that the degree of fearfulness in the morphed faces significantly influenced infants’ looking behavior. In a statistical analysis in which all face pairs were included, we found that infants, regardless of their crawling status, began to show a looking preference for the fearful morphs over the happy face starting from the 60% morph onward as the faces became more fearful. This result confirms previous findings indicating a so-called negativity bias [e.g., 2], or more specifically a fear bias [e.g., 3,4] in the second half of the first year of life. However, this analysis did not show a significant influence of crawling on the infants’ preference scores. We hypothesized that the effect of crawling was relatively weak, since studies with younger infants without any crawling experiences have already shown a fear bias starting from 60% fearful morphs onward [3,4]. Consequently, crawling may have the strongest effect (and thus be more detectable) at or around the 50% fear morph point, as it is the most ambiguous one. We reasoned that in morphs below 50% fear, the fearful expressions may have still been too similar to the happy face to be easily distinguishable from it by either crawling or non-crawling infants. Following the same logic, morphs above 50% may have appeared to be distinct enough from the happy face that both crawlers-and non-crawlers could reliably differentiate between them.

Reviewer 2 Report

The authors report an interesting study on 9- to 10-month-old infants’ processing of emotional faces as a function of their locomotor abilities. Given the lack of research directly investigating the impact of crawling experience on the development of attentional biases towards positive and negative emotional expressions, the authors have the merit of trying to fill a gap in existing literature. Nevertheless, I have several concerns with the introduction and some methodological aspects of the study that I will summarize in the following section.
In general, since there are several grammatical errors throughout the manuscript, I think it would benefit from a careful revision.

INTRODUCTION

  • In the introduction (lines 57-59), the authors report that the transition from the positivity to the negativity bias occurs at the latest at 7 months of age. However, they should refer to the transition from the positivity bias to the fear bias. Indeed, as also mentioned by the authors, several studies demonstrate that the attentional bias for other negative emotions (e.g., anger) appears later in the first year of life when stimuli are presented statically (e.g., Grossmann et al., 2007). In addition, evidence suggests that when 7-month-olds are presented with dynamic emotional faces the transition from the positivity to the negativity bias for angry faces possibly undergoes through a time when both happy and angry facial expressions are perceived as equally salient (Quadrelli et al., 2019).
  • Line 68: “…Kotsoni and colleagues examined 7 month-old infants.”. The authors may want to complete the sentence by specifying what aspects of 7-month-old infants were examined in Kotsoni et al. study.
  • Lines 90-91: “This increased sensitivity seems likely to at least 90 partially stem from adults having far more exposure to such faces over the course of their 91 lives.”. The authors may want to consider adding a reference to justify this sentence.
  • Line 109: “Sorce et al. [18] used such an approach in crawling infants:…”. I believe that the visual cliff can be considered an apparatus or a specific experiment, not an approach.
  • As the authors report in the final part of the introduction section that they collected information concerning stress experience in parents, I expected to find some analyses and results related to this variable and specific hypotheses about expected results linking infants' looking preference and stress levels in parents.

METHODS

  • It seems that the sample size is quite limited, and I wonder if the authors conducted an a priori power analysis to justify the appropriateness of the sample size for the analyses that have been performed. I strongly recommend the authors to provide an estimate of power for their analyses.
  • When describing details about their sample in the participants section, the authors report that they tested 29 infants (18 males and 11 females; 14 crawlers and 15 non-crwalers). I think that it is sufficient to report information about the number of females or males and crawlers or non-crawlers.
  • As the authors report that they collected data about the exact time of the infants’ onset of crawling, I expected to find more information concerning crawling experience (e.g., Mean and SD for the amount of time since infants first started crawling to testing day) in the participants section.
  • When describing their stimulus materials (Lines 189-190), the authors state that “the animations depicted the formation of each morph within a span of 500ms from a neutral facial expression.”. However, I did not understand if animations resulted from the gradual and continuous unfolding from neutral to the emotion of interest or from the consecutive presentation of two frames only. Can the authors clarify this aspect? Also, did the authors measure the quantity of movement and luminance for each animation (e.g., neutral to 100% happy vs neutral to 100% fearful)? I suggest them to perform specific analyses (see for example Grossmann & Jessen, 2017) to exclude that observed results are not due to potential differences in low level perceptual features of the stimuli.
  • I would appreciate if the authors could specify which scales or subscales of the Bayley Scales were used in the current study (lines 205-206)
  • To improve clarity, it would be better if the authors reported the diagonal size of the monitor instead of height and width (line 212).
  • Did the authors consider the possibility to select more specific AOIs instead of simply defining two big rectangles comprising the entire face? I do not see the advantage of using an eye-tracker instead of an experimenter coding infants' looking behavior. I believe that selecting more precisely the AOIs would be more informative.
  • Lines 224-225: how many infants were tested with a 2 point calibration? Why did the authors decide to include these participants anyway?
  • Lines 246-247: “ After the main experimental task, caregivers were asked to fill out the questionnaires.”. Please provide some description concerning the variables obtained from the questionnaires (e.g., Bayley scores: M, SD; Stress level: M, SD) in order to better characterize the sample.

RESULTS

  • Lines 259-262: The authors should better describe the variables included in the preliminary analyses. I believe that by reporting and providing a more detailed description of the data deriving from questionnaires and scales in the methods section will surely help in this sense.
  • Lines 276-279: I found difficult to understand why the authors explored the "morph level" main effect by comparing the preference scores averaged within 10, 20, 30 and 40% levels vs the 50, 60, 70, 80, 90, and 100% levels. Is there a specific reason to split the morph levels in this way? I think it would be more correct if the authors provided post-hoc comparisons between each morph levels. Alternatively, if they have specific theoretical reasons to justify the averaging I suggest them to split the grouping equally (i.e., 10% to 50% vs 60 to 100%).
  • Lines 308-309: when exploring the significant morph level X crawling status interaction the authors provide post-hoc analyses within each morph level only. I would appreciate if the authors report planned comparisons not only within each morph level but also between morph levels within crawling status (e.g., crawlers 40% vs crawlers 50%).
  • I am also curious to know whether the authors performed the main analysis by using the amount of crawling experience as a continuous covariate in their main analyses. I believe this will provide more precise information concerning the role of experience in shaping attentional biases towards emotional faces in infancy.

As an additional minor comment I believe the authors should check their paragraphs and highlight their titles appropriately (e.g., Line 44: Perception and processing of emotional expressions in infancy; Line 176: Emotion task; Line 194: Parents’ stress level; etc..)

Author Response

The authors report an interesting study on 9- to 10-month-old infants’ processing of emotional faces as a function of their locomotor abilities. Given the lack of research directly investigating the impact of crawling experience on the development of attentional biases towards positive and negative emotional expressions, the authors have the merit of trying to fill a gap in existing literature. Nevertheless, I have several concerns with the introduction and some methodological aspects of the study that I will summarize in the following section.
In general, since there are several grammatical errors throughout the manuscript, I think it would benefit from a careful revision.
ï‚· Regarding the comments about our manuscript needing to be revised by native English speaker, one of our native English-speaking colleagues will do a final language-correction pass over the entire manuscript once our revisions for the reviewers are approved.
INTRODUCTION
ï‚· In the introduction (lines 57-59), the authors report that the transition from the positivity to the negativity bias occurs at the latest at 7 months of age. However, they should refer to the transition from the positivity bias to the fear bias. Indeed, as also mentioned by the authors, several studies demonstrate that the attentional bias for other negative emotions (e.g., anger) appears later in the first year of life when stimuli are presented statically (e.g., Grossmann et al., 2007). In addition, evidence suggests that when 7-month-olds are presented with dynamic emotional faces the transition from the positivity to the negativity
bias for angry faces possibly undergoes through a time when both happy and angry facial expressions are perceived as equally salient (Quadrelli et al., 2019).
o Thank you for raising this point about the general negativity bias vs the fear bias. We have now replaced negativity bias with fear bias at appropriate places throughout the manuscript.
o Also thank you for mentioning these studies. We added them to the introduction (see line 59 to 67 in the manuscript) :
ï‚§ In a study by Quadrelli et al. [14], 7-month-olds showed a stronger neural response to happy faces compared to angry ones during static presentation, and a comparable neural response to angry and happy faces when the stimuli were presented dynamically. However, when happy and angry facial expressions were presented statically, in a study by Grossmann et al. [15], 7-month- old infants showed a higher sensitivity to happy than to angry faces.
ï‚· Line 68: “…Kotsoni and colleagues examined 7 month-old infants.”. The authors may want to complete the sentence by specifying what aspects of 7-month-old infants were examined in Kotsoni et al. study.
o Thank you for the advice, I added the bold part for clarity. See line 76 to 77 in the manuscript.
ï‚§ To investigate how fearful an expression needs to be in order for infants to detect the transition from a happy to a fearful facial expression, Kotsoni and colleagues [3] examined 7-month-old infants using a preferential looking task.
ï‚· Lines 90-91: “This increased sensitivity seems likely to at least 90 partially stem from adults having far more exposure to such faces over the course of their 91 lives.”. The authors may want to consider adding a reference to justify this sentence.
o Thanks for mentioning this point. We changed it in the manuscripts (line 92 to 96) as follows:
ï‚§ This increased sensitivity seems likely to at least partially stem from adults having more experiences with such faces over the course of their lives, which makes sense given that the identification of fear or anger in facial expression can be especially relevant for survival. For example, studies have shown that adults’ attention was better captured by negative faces than by positive or neutral ones [16,17].
ï‚· Line 109: “Sorce et al. [18] used such an approach in crawling infants:…”. I believe that the visual cliff can be considered an apparatus or a specific experiment, not an approach.
o We changed it as follows (see line 115):
ï‚§ Sorce et al. [22] used such an apparatus to study crawling infants:…
ï‚· As the authors report in the final part of the introduction section that they collected information concerning stress experience in parents, I expected to find some analyses and results related to this variable and specific hypotheses about expected results linking infants' looking preference and stress levels in parents.
o Thank you for mentioning this point. We added hypotheses, analyses and some discussion.
o Introduction: The current study (see line 166 to 170)
ï‚§ Since the onset of crawling brings many changes in social interaction [5], and caregivers begin to show more negative emotions towards their infants [6,7], we expect that parents of crawling infants have a higher stress level than parents of non-crawling infants. Furthermore, we expect that infants of parents with higher stress levels may show a higher preference for fearful faces compared to infants of parents with lower stress levels.
o Results: (see line 359 to 382)
o Three last repeated measure ANOVAs were run regarding the influence of parents’ stress and recovery levels on infants’ transition to a looking preference for fearful faces. Looking preference scores served as the dependent variable in all three analyses. In the first analy-sis morph level (10% to 100% fear) was a within-subject factor, with the EBF-stress-group (more stressed, n = 15 vs. less stressed n = 14; median split) was a between-subjects factor, and the EBF-overall-stress-score served as continuous covariate. There were no significant main effects or significant interactions (all ps ≥.162). The second analysis was done with morph levels (10% to 100% fearful facial expression) as a within-subject factor, EBF-recovery-group (more recovered, n = 16 vs. less recovered n = 13; median split) as a between-subjects factor, and the EBF-overall-recovery-score as continuous covariate. This analysis did not show any significant results (all ps ≥.102). Thus, neither parents’ stress level, nor parents’ recovery level had an influence on infants’ preference for fearful facial expressions. The third repeated measure ANOVA was also done with morph level (10% to 100% fear) as within-subject factor, the Bf-SR-group (more stressed, n = 16 vs. less stressed n = 13; median split) as between-subjects factor, and the Bf-SR-T-score as a continuous co-variate. Once again, there were no significant main effects or significant interactions (all ps ≥.102). As with the EBF-overall-stress results, also here parents’ stress score was not relat-ed to infants’ looking preference score to the morphs. To get more insight in parents’ stress level, a last analyses about the interpretation of parents’ stress level based on the interpre-tation of the Bf-SR [33]. Based on the Bf-SR-T-scores parents were assigned to the four groups (strikingly low: n = 1; normal: n = 23; slightly elevated: n = 2; markedly elevated: n =1). A χ²-test was run to check the groups regarding their equal distribution. As expected by the group sizes, the groups were not equal distributed, χ²(3) = 45.897, p < .001. Most parents of the current study showed a normal stress level.
o Discussion: general section (see line 456 to 461)
ï‚§ Nevertheless, in our study the parents of crawling infants did not significantly differ from the parents of non-crawling infants regarding their stress level or recovery level. Furthermore, the parents’ stress and recovery levels had no significant influence on infants’ looking
behavior in our emotion task, regardless of the infants’ crawling status.
o Discussion: Limitations and future research (see line 498 to 502)
ï‚§ Another question that we could not reliably answer in our study is the influence of parental stress level on the infants' sensitivity to fearful faces, since the current study involved almost exclusively (88.5%) parents with normal stress levels. Future studies could address this point with a broader selection of study participants, which could be especially valuable from a clinical standpoint.
METHODS
ï‚· It seems that the sample size is quite limited, and I wonder if the authors conducted an a priori power analysis to justify the appropriateness of the sample size for the analyses that have been performed. I strongly recommend the authors to provide an estimate of power for their analyses.
o Thank you for your recommendation. We added a post hoc power analysis as follows (see line 303 to 304):
ï‚§ The post-hoc power analyses revealed a power of .99.
ï‚· When describing details about their sample in the participants section, the authors report that they tested 29 infants (18 males and 11 females; 14 crawlers and 15 non-crwalers). I think that it is sufficient to report information about the number of females or males and crawlers or non-crawlers.
o Changed to (see line 179 to 182): The final sample consisted of 29 healthy full-term infants (8 female and 6 male crawlers [crawling duration: M = 71.93 days, SD = 14.96 days]; 3 female and 12 male non-crawlers [crawling duration: M = 7.67 days, SD = 11.01 days]) with a mean age of 9.90 months (SD = 0.49 months).
ï‚· As the authors report that they collected data about the exact time of the infants’ onset of crawling, I expected to find more information concerning crawling experience (e.g., Mean and SD for the amount of time since infants first started crawling to testing day) in the participants section.
o Please see above.
ï‚· When describing their stimulus materials (Lines 189-190), the authors state that “the animations depicted the formation of each morph within a span of 500ms from a neutral facial expression.”. However, I did not understand if animations resulted from the gradual and continuous unfolding from neutral to the emotion of interest or from the consecutive presentation of two frames only. Can the authors clarify this aspect?
o Thank you for mentioning this point. We added a more detailed explanation to clarify this point. See line 205 to 207.
ï‚§ The animations depicted a continuous formation of each morphed face, and 100% happy face over 500ms (15 frames) from a neutral facial expression (for example, see Figure 2).
ï‚· Also, did the authors measure the quantity of movement and luminance for each animation (e.g., neutral to 100% happy vs neutral to 100% fearful)? I suggest them to perform specific analyses (see for example Grossmann & Jessen, 2017) to exclude that observed results are not due to potential differences in low level perceptual features of the stimuli.
o Thank you for raising this point. Since our stimuli were presented simultaneously, and both images started from a neutral facial expression (and the images were in motion for the same amount of time) we do not think that differences in motion time were responsible for the looking preferences given that all infants saw the same contrasts, and our main research question regarding crawling was addressed with a between-subjects approach.
o Since we think this point is important to note, we added it to our discussion. See line 462 to 473.
ï‚§ An important point to note is our use of dynamic stimuli, which can introduce low-level motion-related differences between the visual stimuli. This question is quite important, since studies have shown, infants’ higher attention to moving stimuli compared to static stimuli [e.g., 35]. This point was raised by Grossmann and Jessen [36] in reference to another study [13] which used dynamic stimuli. Grossmann and Jessen [36] noted that in this study [13] the fearful facial expressions contained more movement than the happy and neutral facial expressions. However, the happy animated face in our study was only used as a contrast against which to measure the infants’ sensitivity to the fearful morphs, which was the real variable of interest. Furthermore, to test our hypothesis regarding the effects of crawling, we used a between-subjects design where all infants saw the same selection of stimuli, and thus any low-level visual differences between the happy and fearful faces are very unlikely to significantly impact our findings.
o We did not adjust our stimuli for luminance. It may be worth trying in future experiments, although this might result in slightly less “lifelike” stimuli.
ï‚· I would appreciate if the authors could specify which scales or subscales of the Bayley Scales were used in the current study (lines 205-206)
o We used the crawling item of the Bayley scales. See line 228 to 232.
ï‚§ Infants’ crawling status: To determine the infants’ crawling status, parents were asked regarding their infants’ crawling status based on the definition of crawling of the BLINDED FOR REVIEW version of The Bayley Scales of Infants and Toddler Development-III [25]. Crawling was defined as follows: “Child makes forward progress of at least 1.5m by crawling on hands and knees”.
ï‚· To improve clarity, it would be better if the authors reported the diagonal size of the monitor instead of height and width (line 212).
o We changed it to the diagonal size instead of height and width. See line 237 to 238.
ï‚§ Emotion task: Stimuli were presented using E-Prime [32] on an LCD monitor (diagonal size: 61 cm) with a resolution of 1920 x 1080 pixels.
ï‚· Did the authors consider the possibility to select more specific AOIs instead of simply defining two big rectangles comprising the entire face? I do not see the advantage of using an eye-tracker instead of an experimenter coding infants' looking behavior. I believe that selecting more precisely the AOIs would be more informative.
o We decided to use AOIs which covered the entirety of the facial stimuli because we were interested in the infants’ perception of the facial expressions as a whole rather than the influence of specific parts of the face. However, this could be an interesting question for future studies.
ï‚· Lines 224-225: how many infants were tested with a 2 point calibration? Why did the authors decide to include these participants anyway?
o We decided to also include infants with a two point calibration because we examined large AOIs covering the entirety of the face stimuli and not smaller AOIs, and thus a 2-point-calibration was still sufficient in our view.
ï‚· Lines 246-247: “ After the main experimental task, caregivers were asked to fill out the questionnaires.”. Please provide some description concerning the variables obtained from the questionnaires (e.g., Bayley scores: M, SD; Stress level: M, SD) in order to better characterize the sample.
o Since we did analyses regarding possible differences between crawlers and non-crawler, we added a table after the first paragraph in the results section. See line 293:
Table 1
Means and standard deviations of the different questionnaires
Questionnaires’ variables
Crawlers (n = 14)
Non-crawlers (n = 15)
Number of siblings
M = 0.64; SD = 0.33
M = 0.40; SD = 0.63
Mothers’ educational level
M = 3.38; SD = 0.87
M = 3.27; SD = 0.88
Fathers’ educational level
M = 2.92; SD = 1.38
M = 3.13; SD = 0.99
Infants’ social-emotional age
M = 1.18; SD = 0.21
M = 1.08; SD = 0.24
Parents’ stress level
(Bf-SR-T-score)
M = 49.64; SD = 6.97
M = 51.47; SD = 7.99
Parents’ overall stress
and recovery level
(EBF)
M = 1.56; SD = 0.59
M = 2.82; SD = 0.79
M = 1.42; SD = 0.61
M = 2.78; SD = 0.70
Note. Mothers’ and fathers’ educational level ranged from 0 (no degree) to 5 (promotion/habilitation). For one infant of the crawling group, the parents did not answer the questions regarding the mother’s and father’s educational level.
RESULTS
ï‚· Lines 259-262: The authors should better describe the variables included in the preliminary analyses. I believe that by reporting and providing a more detailed description of the data
deriving from questionnaires and scales in the methods section will surely help in this sense.
o We added a more detailed description of the questionnaires to the methods section. See line 213 to 232:
Parents’ stress level: In order to obtain information about the current mental state of the parents, we used a state of mind questionnaire [33]. The state of mind questionnaire includes 24 items about the current well-being of the respondent. The raw values obtained are converted into T-values (or PRs, or stanine values) for subsequent interpretation. Thus, a T-score > 60 is considered slightly elevated, a T-score above ≥ 63 is considered moderately elevated, and a T-score ≥ 70 is considered significantly elevated. In the opposite case, T values ≤ 40 are considered very low [33]. Additionally, the BLINDED FOR REVIEW of the Recovery-stress-questionnaire [34] was used to get information regarding parents’ current extent of recovery and stress over the last three days and nights. The basic version with 24 items was used [34].
Infants’ social-emotional development level: To assess the infants' general level of social-emotional development, we used age-specific emotion related items from a social-emotion questionnaire [35] administered to the parents. The questions referred to self-image, emotional independence, awareness of reality, moral development, anxiety, impulse control, and regulation of emotions.
Infants’ crawling status: To determine the infants’ crawling status, parents were asked regarding their infants’ crawling status based on the definition of crawling of the BLINDED FOR REVIEW version of The Bayley Scales of Infants and Toddler Development-III [25]. Crawling was defined as follows: “Child makes forward progress of at least 1.5m by crawling on hands and knees”.
ï‚· Lines 276-279: I found difficult to understand why the authors explored the "morph level" main effect by comparing the preference scores averaged within 10, 20, 30 and 40% levels vs the 50, 60, 70, 80, 90, and 100% levels. Is there a specific reason to split the morph levels in this way? I think it would be more correct if the authors provided post-hoc comparisons between each morph levels. Alternatively, if they have specific theoretical reasons to justify the averaging I suggest them to split the grouping equally (i.e., 10% to 50% vs 60 to 100%).
o Thank you for raising this point. When we wrote the manuscript, we decided to split the halves that way (10% to 40% vs 50% to 60%) since in view of the results of our first analyses, there was a transition regarding fear preference starting from the 50% fearful morph. Based on your advice regarding an equal split of the halves, we decided to compare the halves in the way you mentioned (10% to 50% vs. 60% to 100%). See line 307 to 320.
ï‚§ In order to further analyze the increase of the preference scores with increasing morph level from the previous analysis (see Fig. 4), we compared the average preference scores at morph levels 10% to 50% (the first half of the continuum) against the average preference scores at morph levels from 60% to 100% (second half). This analysis also allowed us to examine whether the infants in the current study also show a general fear bias as shown by previous studies which investigated similar age groups of infants using
fearful and happy facial stimuli [3,4]. We again ran a repeated-measures ANOVA on the preference scores with the morphing degree (10% to 50% vs. 60% to 100%, as described above) as a within-subject factor and crawling status (crawlers vs. non-crawlers) as a between-subjects factor. Results showed a significant effect of the morphing degree, F(1, 27) = 7.797, p = .009, η2part = .224, with infants showing higher preference scores for morphs from the second half of the continuum (Figure 5). There was no significant interaction between the morphing degree and crawling status, F(1, 27) = 2.092, p = .160, η2part = .072, and no main effect for crawling status, F(1, 27) = 0.372, p = .547, η2part = .014.
ï‚§
ï‚· Lines 308-309: when exploring the significant morph level X crawling status interaction the authors provide post-hoc analyses within each morph level only. I would appreciate if the authors report planned comparisons not only within each morph level but also between morph levels within crawling status (e.g., crawlers 40% vs crawlers 50%).
o We structured this post-hoc analysis in this way because it allowed us to directly contrast the preference scores of the crawlers versus the non-crawlers in order to evaluate the effect of crawling experience which is our main research interest in this study.
ï‚· I am also curious to know whether the authors performed the main analysis by using the amount of crawling experience as a continuous covariate in their main analyses. I believe this will provide more precise information concerning the role of experience in shaping attentional biases towards emotional faces in infancy.
o Thank you for mentioning this point. We had thought of doing such an analysis, but it was unworkable because many infants in our non-crawling group had 0 days of crawling experience and very few that had some intermediary number of days of
crawling experience. Thus, we believe that the jump from these 0 experience crawlers to the infants who actually had some crawling experience would make the days of crawling experience a poor continuous variable for our particular set of data.
As an additional minor comment I believe the authors should check their paragraphs and highlight their titles appropriately (e.g., Line 44: Perception and processing of emotional expressions in infancy; Line 176: Emotion task; Line 194: Parents’ stress level; etc..)
ï‚· Thank you for this suggestion. We bolded the paragraphs to highlight them.

Round 2

Reviewer 2 Report
The authors addressed all raised points.

Author Response
Thank you for your kind reply. Our colleague has now completed the language editing, and we now feel that the manuscript is ready.